# An Analysis of Preference Weights and Setting Priorities by Irrigation Advisory Services Users Based on the Analytic Hierarchy Process

Itzel Inti Maria Donati [1], Davide Viaggi [2], Zorica Srdjevic [3], Bojan Srdjevic [3], Antonella Di Fonzo [4,*], Teresa Del Giudice [5], Orlando Cimino [4], Andrea Martelli [4], Anna Dalla Marta [6], Roberto Henke [4] and Filiberto Altobelli [4,*]

[1] Department of Architecture and Design Landscape and Environment Sapienza, University of Rome, Piazza Borghese 9, 00186 Rome, Italy; itzelinti.donati@uniroma1.it

[2] Department of Agricultural and Food Sciences, Alma Mater Studiorum Università di Bologna, Viale Fanin 50, 40127 Bologna, Italy; davide.viaggi@unibo.it

[3] Department of Water Management, Faculty of Agriculture, University of Novi Sad, Trg D. Obradovica 8, 21000 Novi Sad, Serbia; zorica.srdjevic@polj.uns.ac.rs (Z.S.); bojan.srdjevic@polj.uns.ac.rs (B.S.)

[4] Council of Research in Agriculture and Analysis of Agricultural Economics, Research Centre for Agricultural Policies and Bio-Economy, Via Barberini 36, 00187 Rome, Italy; orlando.cimino@crea.gov.it (O.C.); andrea.martelli@crea.gov.it (A.M.); roberto.henke@crea.gov.it (R.H.)

[5] Department of Agricultural Sciences, University of Naples Federico II, Reggia di Portici, Via Università 100, 80055 Naples, Italy; teresa.delgiudice@unina.it

[6] Department of Agrifood Production and Environmental Sciences, University of Florence, Via delle Cascine 5, 50144 Florence, Italy; anna.dallamarta@unifi.it

* Correspondence: antonella.difonzo@crea.gov.it (A.D.F.); filiberto.altobelli@crea.gov.it (F.A.)

**Abstract:** *Objective:* Stakeholders—farmers from four different European areas (Campania (IT), Kujawsko-Pomorskie (PL), Limburg (NL), Andalusia (ES))—are asked to share, from the OPERA project, their opinions on five criteria that all aim at improving the use of irrigation advisory services (IASs). Each criterion has different characteristics that affect the way farmers rank it. The present study has two objectives. The first is to individuate the priorities of the preferences expressed by the stakeholders. The second objective is to carry out a ranking of the weights of the criteria by case study, ranking the groups and their associated properties among farmers' profiles. *Methods:* The answers to 120 questionnaires dispensed to the future users of IASs in the four agricultural sites were analyzed in detail, and then the given priorities were evaluated through the analytic hierarchy process (AHP). The AHP methodology was used to determine the relative weights of the five assessment criteria, and finally, to select the one with major value. *Results and conclusions:* The results show that A5 (assuring economic sustainability) was the most important criterion. The contributions provided by this study are twofold: Firstly, it presents an application of a methodology that involves the conversion of a linguistic judgement of farmers in a correspondence weight. Secondly, it tackles decision making regarding improving the use of IASs, evaluating the preferences expressed by the stakeholders. Irrigation advisory services can play a key role in assisting users to adopt new techniques and technologies for more efficient water use and increased production.

**Keywords:** multicriteria decision analysis; AHP; irrigation advisory services; agricultural decision making; economic sustainability

## 1. Introduction

The interest in promoting a form of agriculture capable of adapting to climate change has made the management of water resources one of the key points in the reform of the CAP 2023–2027 [1]. As a matter of fact, "*Foster sustainable development and efficient management of natural resources such as water, soil, air*" and "*Fostering knowledge, innovation and*

*digitalisation in agriculture"* have been designated as two of the ten new strategic objectives of the new CAP 2023–2027. In order to pursue more efficient and sustainable water use, EU countries are called, among other things, to encourage research and innovation in the sector by the implementation of "smart irrigation" technologies. An efficient use of water for irrigation is a priority driven by the evidence that many areas in the Mediterranean region suffer structural water scarcity, imposed by the periodic droughts and by the expansion of water demands of agriculture and other sectors of society [2]. With the advancement of climate change, higher temperatures, and changing precipitation patterns, the demand for water by the agricultural sector has increased. It has started to affect not only areas where irrigation has always been an essential element of agricultural production (southern Europe) but also areas traditionally considered not irrigated, such as some areas of central and northern Europe. In this context, the EU research project OPERA—"Operationalizing the increase of water use efficiency and resilience in irrigation", http://opendata.waterjpi.eu/dataset/2a2a87e0-5c84-42cd-a9da-ecac0bbb9257/resource/1b07850f-c7e8-4a0d-86c8-180ff3e1bae5/download/d5.1_inception_report_opera.pdf (accessed on 24 July 2023) is a program financed under ERA-NET, which is part of Water JPI. Water JPI aims to tackle the challenge of "achieving sustainable water systems for a sustainable economy in Europe and abroad". Within the context of a sustainable economy, OPERA focuses on the sustainable management of water resources in agriculture and the use of irrigation advisory services (IASs), and thus, intelligent irrigation systems that provide information to a large number of farmers have become useful tools for irrigation programs. The issue is not new, and extensive research and investments have been made to develop more advanced methods and practices to accurately provide water to the crops based on their needs.

Technological advances in IASs continue to increase rapidly [3–9]. Along with it, the behavioral and socio-economic determinants of farmers for the adoption of these efficient irrigation technologies are also evolving. The success of these technologies can be supported by the integration of stakeholders' needs in the design of IASs [10,11].

Nevertheless, the literature on the subject appears to be poor in case studies focused on identifying the needs of farmers, who are the end users of IASs.

There is still a significant disparity between the availability of technologies for efficient water usage and the acceptance of these technologies. One of the reasons is the lack of emphasis on establishing an efficient support system to aid farmers in adopting and effectively operating new techniques and technologies. Through four case studies in the EU context, this paper will address the following questions:

- How can OPERA cope with these issues while taking into consideration the feedback obtained from the stakeholders' answers and making use of the current AHP analysis?
- Are there any spatial differences or correlations among the criterion improvements selected by the stakeholders?
- Which of the criteria seem to have major weight according to the stakeholders?

This study is organized into six sections: Sections 1 and 2 describes the background research, general area of interest, and the topic of focus; Section 3 presents the study areas chosen for the research activity and the data and research methodology; in Sections 4 and 5, the research results and discussions are presented; and, finally, in Section 6, the main conclusions and future research design are reported. Within this framework, the application of AHP demonstrates that a multi-criteria problem can be approached specifically for each case study. Nevertheless, an overall result involving all study areas can be achieved. One of the advantages of the AHP method is to support both individual and group decision-making processes not only with a quantitative but also a qualitative approach. Since the 1990s, there has been a growing number of studies applying the AHP to deal with decision-making problems in agriculture [12–23]. From these works, the utility of the AHP has arisen for understanding heterogeneous farming systems and how farmer behavior is needed for tailoring policy instruments. Against an agricultural water management background, it also helps to share these frontiers for more efficient, equitable, and sustainable outcomes.

## 2. Background

*Irrigation Advisory Services*

Several factors determine the quantity of irrigation water employed in agriculture, ranging from the variety of crops and cultivation approaches to soil properties and the irrigation method, among others. Hence, agriculture itself presents prospects for improved water administration and conservation, encompassing both conventional farming practices and innovative agricultural technologies. Among the latter, irrigation decision support systems (DSSs) can assist farmers in making informed decisions, leading to enhanced profitability by optimizing water usage and ensuring maximum crop yield in a particular growing season. These systems are primarily designed to simulate or forecast crop water requirements, presenting a range of choices. Under this scenario, irrigation advisory services (IASs) are considered a useful DSS to help farmers achieve the best efficiency in irrigation water use and to increase their incomes by obtaining the highest possible crop yield. Irrigation advisory services are a set of activities that aim to provide technical and professional support to farmers and agricultural operators in the management of cropland irrigation. In recent decades, the research has focused on investigating new IASs tools, which has contributed to the evolution of the performance capabilities of the services. Nowadays, IASs can be implemented in a broad range of agricultural situations, and they can easily be combined with several software programs. IASs are able to deal with the following:

- Satellite-based irrigation volumes are able to perform a site-specific evaluation of irrigation volumes, integrating remote sensing data with a geographic information system (GIS) [24]. In some cases, the research has been focused on quantifying several irrigation and drainage performance indicators with the support of a GIS.
- Development delivery data from a desktop application to via the web, considering that the graphical user interface is a key element for the successful use of the services (PlanteInfo, WIESE, IRRINET, BEWARE, ISS-ITAP, IrriSAT, IRRISAT) [25–27].
- Biophysical variables, surface soil water content, and canopy water content; for example, some studies have been inquiring about how to estimate separately determine soil evaporation and crop transpiration [28–30].
- In the context of remote-sensing tools, some studies have been carried out as a part of the project DEMETER (Demonstration of Earth observation technologies in routine irrigation advisory services), which deals with the transmission of personalized irrigation scheduling information to the users, related to an extended period of time (e.g., on past, present, and future weather) [31].
- Some studies have investigated the idea of an IAS tailored to the distinct circumstances of farmers. The findings indicate distinct farmers' inclinations, particularly for obtaining weather predictions from the service and for the characteristics associated with water data registration [32,33].

As indicated above, the research has made notable steps forward, progressing in the technical aspects at the basis of DSS programming for irrigation, and has made the use of IASs more and more efficient. It has also made the use of these tools applicable in various agricultural contexts.

The strengthening of the aspects of the research activity mentioned above deserves to be further investigated to understand the judgment of the end users and their needs, with the aim of favoring the implementation of IASs in the management of water resources in the field.

## 3. Materials and Methods

*3.1. Study Areas*

The analytic hierarchy process (AHP) is a general theory of measurement [34], and it has been used in the present study to analyze the verbal judgments of IASs end users

belonging to four different geographical areas: Italy, Netherlands, Spain, and Poland. The characteristics of these regions are summarized below.

Campania (IT)—For this case study, 40 interviews were collected. The key stakeholders featured in the sample interviews included not only farmers but also representatives from the regional government, land and water reclamation authorities, farmer associations, local policymakers, and legislators. The farm activity is focused on growing silage maize, wheat, tomato, and other vegetables (such as peppers and beans). Additionally, 12% of farmers in the area are engaged in cereal production and livestock rearing. Within the project, the crucial investigation consisted of exploring the potential for adaptation and addressing the needs of the end users, as well as identifying optimal approaches for dynamically forecasting crop water requirements through the assimilation of remote sensing observations and numerical weather predictions in a crop growth model.

In the Kuyavian-Pomeranian province (PL), 53 interviews were recollected. In this case study, two demonstration areas were involved: the first one was located in the Zglowiaczka River catchment in a small productive field, and the second one was located in the Upper Notec River catchment in a big productive field. Vegetables are cultivated as well as irrigated in both fields. Although this is a region of intensive agricultural crop production, in an average year, it suffers concerning rainfall water deficits. The main issues affecting these areas are water shortage in the growing season, the use of clean renewable deep groundwater resources for irrigation, insufficient efficiency of water used for irrigation, and the lack of an irrigation decision support system for farmers based on the current crop water needs and weather forecasts.

Andalucía (ES)—Agriculture in the Mediterranean region is dealing with serious problems related to the present drought and the general scarcity of water resources, resulting in an increasing water demand [35]. These difficulties are expected to worsen due to the future predicted severe water scarcity in the Mediterranean area. Olive cultivation has been chosen as a case study, as it is a crucial economic sector representing 24% of the value of agricultural production in the Andalusia area. It covers an area of approximately 1.5 million hectares (around 17% of the total region's surface, accounting for 60% of the national surface dedicated to olive crops and 30% of the European surface) and contributes to about 40% of global olive oil production and 20% of global table olive production. Additionally, it is a significant source of wealth and employment, supporting over 22 million wages annually. It is also essential for social and territorial cohesion and possesses high environmental value, shaping the Andalusian territory and culture. While some facilities and advice services are available in this area, the current solutions have not been effectively implemented, leading to unsatisfactory results.

Limburg (NL)—Seven interviews were collected in one of the less dry regions of Northern Europe. Among the interviewed farmers, mobile irrigation (overhead sprinkling) systems are in widespread use. Irrigation management is supported by weather forecasts that anticipate crop water availability and by supplementary information through sensors (local or remote). Water availability for irrigation in the Netherlands in previous years was not a frequent concern. However, due to climate change, they encounter more drought periods, and local water boards temporarily proscribe farmers from using surface water and groundwater for irrigation. When this occurs more frequently in the foreseeable future, an enhanced water supply will be necessary.

### 3.2. Data Collection

### 3.2.1. Identifying Respondents' Profiles

As shown in Figure 1, this study was organized into a four-phase methodology. The first step of the investigation was identifying the needs and demands of the users. Each case study partner selected the particular stakeholders on the basis of the "Guidelines for analysis and selection of stakeholders". This approach is based on a "snowball sampling design". Information is available in the "D1.1 Assessment of user requirements of the sector" of OPERA at this link: http://opendata.waterjpi.eu/dataset/2a2a87e0-5c84-42cd-a9da-ecac0bbb9257/resource/

 (accessed on 24 July 2023). Identified and contacted stakeholders were asked to identify further stakeholders, starting with the case study partners. The questionnaires were addressed to a total of 120 farmers (users of IAS), distributed among the following study areas: Campania (IT): 40; Limburg (NL): 7; Kujawsko-Pomorskie (PL): 53; and Andalusia (ES): 20.

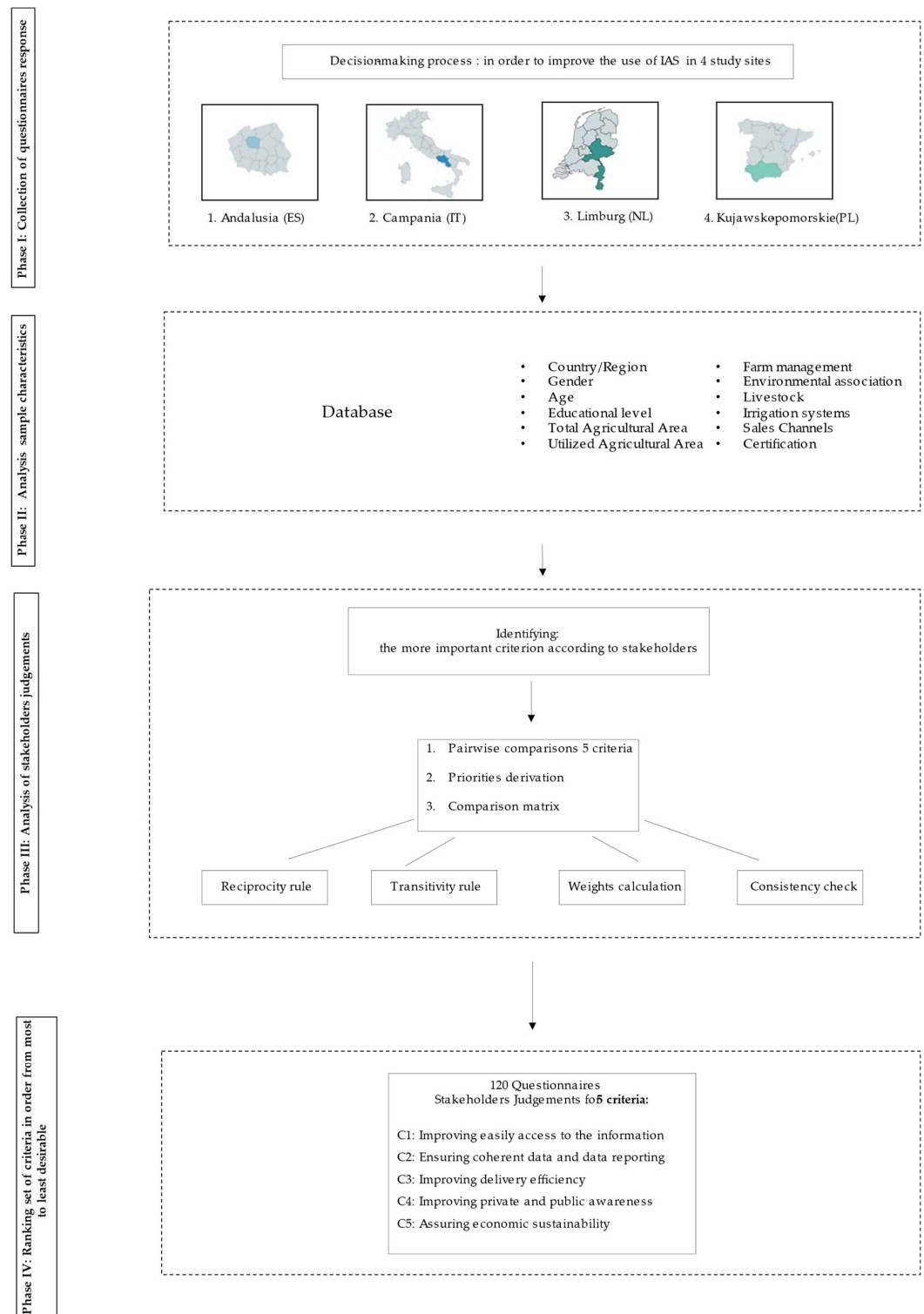

**Figure 1.** Research methodology.

### 3.2.2. Questionnaires

The questionnaire was structured into four sections. The first section was focused on general interview information: activity, gender, age, educational level, farm's location, farm surface, farm management, cultivated crop, main irrigation systems, sales channels, etc. Using this information, a database was developed. Phase two of the research (Figure 1) was possible to realize, combining the outcomes of the AHP and the database analysis (Figure 1). The above-mentioned information was important in order to group the weights of criteria evaluated by the stakeholders and to reveal the inter-relations between the technical factors expressed in the proposed questionnaires and the weights resulting from the employment of the AHP.

Section 2 of the questionnaire was named "Improving water use efficiency and the use of advisory services" and was related to the data that were analyzed by the AHP. The farmers, regarded as stakeholders, expressed their preferences among a set of criteria (Table 1), answering the follow question: "Compare criteria C1 and C5 using Saaty's scale 1–9. According to you, is C1 more important than C2, and by how much?" The pairwise comparison at the core of the AHP methodology was inserted into the questionnaires and is discussed below. Section 3 of the questionnaire, named *"Developing policies and strategy suggestions for improving use of irrigation advisory services at farm level: an Italian experience"* included: (i) four questions related to the internal strengths that farmers could come across when adopting IASs; (ii) the weaknesses that farmers may face when adopting IASs; (iii) the opportunities arising for farmers when adopting IASs; and (iv) the threats to companies when adopting such IASs. Finally, Section 4, named *"Eliciting farmers' individual risk attitude"*, provided suggestions about the adoption of innovative tools proposed by the OPERA project. These tools depend on the farmers' subjective attitudes toward taking risks. The questionnaires were translated from English into the native language of each country.

**Table 1.** Description of AHP criteria.

| Type of Criteria | Description |
|---|---|
| C1. Improving easy access to information | Refers to the ease of access to information for farmers, either through electronic information (SMS, email, etc.), more traditional communication systems, technical operators and journals, newspapers, etc. |
| C2. Ensuring coherent data and data reporting. | Refers to the ability to implement an IAS based on high-quality data, providing valuable technical information to farmers. |
| C3. Improving delivery efficiency | Refers to the ability to ensure prompt and constant delivery of information to farmers. |
| C4. Improving private and public awareness | Refers to improving public awareness and preparedness by informing the public about the risks and consequences in case of excessive use of water for irrigation related to environmental and economic phenomena (e.g., water scarcity, conflict over use of water with others economic sectors). |
| C5. Assuring economic sustainability | Refers to the cost of IAS, which should be economically justified (i.e., economically affordable). |

### 3.3. Multi-Criteria Decision Analysis—Selection of the Analytical Hierarchy Process (AHP)

In the agricultural sector, the main problems that multi-criteria decision analysis (MCDA) is facing are related to their evolution in terms of technological progress (equipment, fertilizers, pesticides, new plant varieties, irrigation systems).

This use of new production systems has been associated with an increase in the size and degree of specialization of agricultural operations [36]. In the literature, several approaches are proposed in the agricultural sector to assist decision makers, such as farmers and their associations, policy makers, and local and regional authorities, to efficiently explore a range of criterion farm management practices, and thus, identify pathways toward sustainability.

When criteria need to be classified into ordered classes, a sorting method has to be applied, but much less attention has been paid to investigating this kind of problem, especially in the case of multiple decision-makers asked to give subjective scores to different criteria based on qualitative criteria.

The analytic hierarchy process (AHP) is an MCDA developed by Thomas L. Saaty in the 1970s [34].

Considering the number of MCDA methods available (PROMETHEE, MACBETH, ELECTRE, TOPSIS), as suggested by the literature [37–39], there are several methods to choose an appropriate MCDA.

The decision to use the AHP in this work was guided by a series of drivers/reasons, summarized as follows:

- Ratio scale and pairwise comparison: The fundamental process involves the comparison of two stimuli, which are also referred to as alternatives, under a particular criterion or two criteria. The decision maker was asked to determine if they were indifferent towards the two stimuli or if they had a weak, strict, strong, or very strong preference for one of them. Understanding this structure is more intuitive for the respondent and facilitates stakeholder participation. The criteria analyzed in this study were identified within the OPERA project, for which detailed information can be found at the following link: http://opendata.waterjpi.eu/dataset/opera-operationalizing-the-increase-of-water-use-efficiency-and-resilience-in-irrigation (accessed on 24 July 2023).
- *Stakeholders*: The AHP can support complex decisions in which several stakeholders are involved, as in the case of the present study. The construction of the database (areas, farm management, irrigation systems) demonstrates that different interest groups are implicated [40].
- *Software*: The AHP is one of the most popular MCDA methods and is backed by a large variety of software offering diverse data management and representation capabilities [41].

### 3.4. Application of Analytic Hierarchy Process

The AHP comprises three principal operations, including hierarchy construction, priority examination, and consistency analysis. In the present study, these steps were carried out as shown in Figure 2 [42].

As mentioned above, the objective of the mathematical procedure is to estimate the weights of five criteria from a matrix of pairwise comparisons $A(a_{ij})$) generated following both the transitivity rule and the reciprocity rule. The reciprocal condition or Axiom 1 defines that given two criteria $(A_i, A_j) \in A \times A$, the intensity of preference of $A_i$ over $A_j$ is inversely related to the intensity of preference of $A_j$ over $A_i$.

Transitivity rule is:

$$a_{ij} = a_{ik} \cdot a_{kj} \tag{1}$$

Reciprocity rule is:

$$a_{ij} = \frac{1}{a_{ji}} \tag{2}$$

where $a_{ij}$ is the comparison of criteria $i$ and $j$.

If we suppose that preferences (weights) $p_i$ are known, a perfectly consistent matrix can be constructed because all of the comparisons $a_{ij}$ satisfy equality:

If the preferences (weights) $p_i$

$$a_{ij} = \frac{p_i}{p_j} \tag{3}$$

where $p_i$ is the priority of the alternative $i$, and the completely consistent matrix is:

$$A = \begin{bmatrix} p_1/p_1 & \cdots & p_1/p_n \\ \cdots & 1 & \cdots \\ p_n/p_1 & \cdots & 1 \end{bmatrix} \tag{4}$$

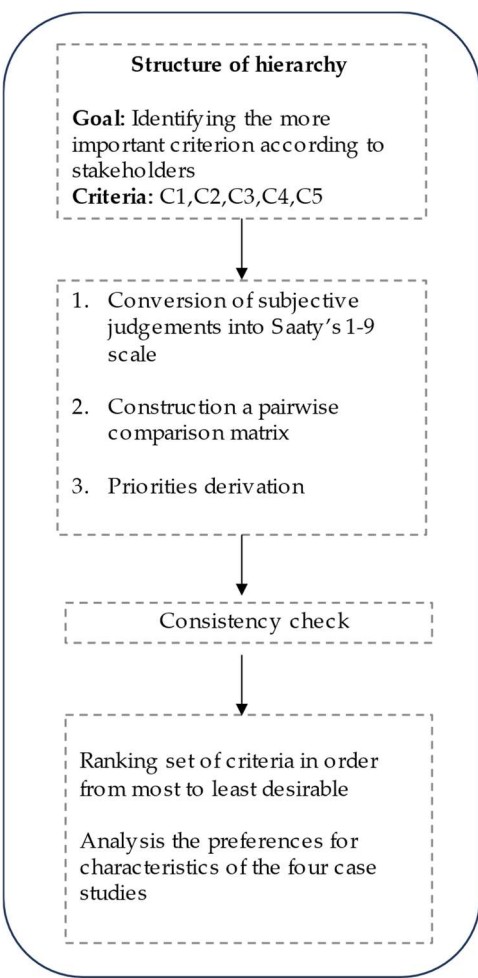

**Figure 2.** AHP steps in the present study.

We can apply the following formula from the matrix above:

$$\sum_j \frac{p_i}{p_j} \cdot p_j = n p_i \tag{5}$$

Thus, the product of row $i$ by the priority vector $p$ gives $n$ times the priority $p_i$. By multiplying all the elements of the comparison matrix $A$ by the priority vector $p$, the following equality is obtained:

$$A = \begin{bmatrix} p_1/p_1 & \cdots & p_1/p_n \\ \cdots & 1 & \cdots \\ p_n/p_1 & \cdots & 1 \end{bmatrix} \begin{bmatrix} p_1 \\ \cdots \\ p_n \end{bmatrix} = n \begin{bmatrix} p_1 \\ \cdots \\ p_n \end{bmatrix} \tag{6}$$

The priorities (weights) of the compared criteria are not known in advance. As priorities only make sense if derived from consistent or near-consistent matrices, a consistency check must be applied. Several methods have been proposed to measure consistency [43,44] Also, normalization plays a key role in obtaining meaningful results from AHP analysis.

The functioning of the model is strictly related to pairwise comparison, involving the level of criteria in the present study. The result of the pairwise comparison is expressed by Saaty's scale. The function in detail is as follows:

First, two criteria—A and B—are compared using the numerical scale ranging from 1 to 1/9 (Saaty's scale), where number 1 means both criteria have the same importance, and they are equal; number 9 means criterion A is 9 times more important than criterion B; 1/9 means that criterion B is nine times more important than criterion A.

Second, it is necessary to compare all elements pairwise with respect to the objective. In the following step, the comparisons are arranged in a matrix. From this matrix, the computed weights for the different criteria are generated.

In 1977, Saaty [45] proposed the consistency ratio (*CR*) to measure the reliability of information contained in a pairwise comparison matrix:

$$CR = \frac{CI}{RI}$$

This is a ratio of the consistency index (*CI*) and random index (*RI*), and it is given by Relation (7)

$$CI = \frac{\lambda_{max} - n}{n - 1}, \tag{7}$$

where $\lambda_{max}$ is computed as

$$\lambda_{max} = \frac{\left(\sum_{j=1} a_{1j}\, p_j\right)}{p_1}$$

The *CR* provides a way of measuring how many errors were created when providing the judgments; a rule-of-thumb is that if the *CR* is below 0.1, the errors are fairly small and thus, the final estimate can be accepted. The first step for computing the *CR* is determining the eigenvalue ($\lambda m$), followed by determining the *CI* [46].

## 4. Results

The AHP was applied to guide a decision-making process, with the ultimate goal of improving the use of IAS among farmers. The first results of the study demonstrate that the most common decision was *Assuring economic sustainability* (C5), as shown in Table 2.

**Table 2.** Overall results of the four study areas.

|  | Criteria | Weights of Criteria | Final Ranking |
|---|---|---|---|
| Evaluating Possible Adoption Options of IAS | C1: Improving easy access to information | 0.207 | 3 |
|  | C2: Ensuring coherent data and data reporting | 0.218 | 2 |
|  | C3: Improving delivery efficiency | 0.196 | 4 |
|  | C4: Improving private and public awareness | 0.148 | 5 |
|  | C5: Assuring economic sustainability | 0.231 | 1 |

There was heterogeneity among the farmers' judgments, which involved a clear difference in weights between the most important criterion and the criterion with a lower weight. By looking at the distribution of the priority values (Table 2), the weights vary, with a minimum weight of 0.15, attributed to criterion C4—*Improving private and public awareness*, which refers to improving public awareness and preparedness by informing the public about the risks and consequences in case of the excessive use of water for irrigation related to environmental and economic phenomena e.g., water scarcity, conflict over the use of water with other economic sectors. A maximum weight of 0.23 was attributed to criterion C5—*Assuring economic sustainability*.

In the following stage, the weights were grouped according to the decision makers' profiles. The next step was to relate the preferences expressed by the stakeholders and to analyze the key information provided by the interviews. Subsequently, a mathematical

aggregate of the weights of each criterion was calculated using the geometric mean method. While this section is not intended to be an exhaustive account of all results of this work, it aims to provide a broad picture of the most relevant results for each pilot area.

Campania (IT): The results show that the priority rankings of the group are quite "flattened", which may be partly due to the inconsistencies among the elements of the pairwise comparison matrix (hinting at some randomness in the answers). Table 3 shows that C5—*Assuring economic sustainability* and C4—*Improving private and public awareness* are the most preferred options. The proposed grouping procedure can be used to discuss some observations. The criteria weights of the Italian study site were compared to the results obtained from the aggregate weight of the Netherlands pilot area. In both areas, (Figures 3 and 4) farmers rear livestock as one of the main farm activities, which involves growing feed crops. Among these samples, the farmers who grew grassland and ryegrass preferred criterion C5—*Assuring economic sustainability*.

**Table 3.** Weights and ranks of criteria in the four study sites.

| Criteria | Andalusia (ES) | | Campania (IT) | | Kujawsko-Pomorskie (PL) | | Limburg (NL) | |
|---|---|---|---|---|---|---|---|---|
| | Weights | Final Ranking | Weights | Final Ranking | Weights | Final Ranking | Weights | Final Ranking |
| C1 | 0.194 | 3 | 0.194 | 3 | 0.163 | 5 | 0.233 | 1 |
| C2 | 0.177 | 4 | 0.177 | 4 | 0.209 | 3 | 0.194 | 3 |
| C3 | 0.126 | 5 | 0.126 | 5 | 0.222 | 1 | 0.222 | 2 |
| C4 | 0.196 | 2 | 0.196 | 2 | 0.185 | 4 | 0.089 | 5 |
| C5 | 0.296 | 1 | 0.296 | 1 | 0.206 | 2 | 0.182 | 4 |

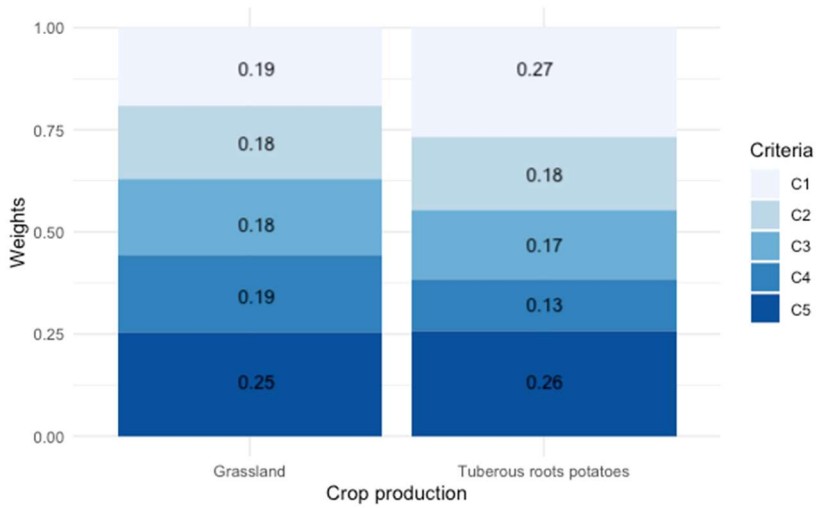

**Figure 3.** Weights of criteria grouped according to the crop production in the Netherlands. C1—Improve easy access to information; C2—Ensure coherent data and data reporting; C3—Improve delivery efficiency; C4—Improve public and private awareness; C5—Ensure economic sustainability.

Limburg (NL): The preference of the farms that have adopted surface irrigation was C5—*Assuring economic sustainability*. As Walker argues, in his study published by FAO in 1980 [47], one of the advantages of surface irrigation is that these systems are inexpensive to develop at the farm level. The control and regulation structures are simple, durable, and easily constructed with cheap and readily available materials. The survey illustrates the financial aspect, which is an important issue to be considered for designing and developing water management strategies for farmers belonging to this profile. As shown in Figure 5, farms that use sprinkler irrigation identified C1—*Improve easy access to information* as more important. Among the three types of irrigation systems, these are the most sensitive to

weather conditions. For example, strong winds can affect the efficiency of the spraying of water from sprinklers.

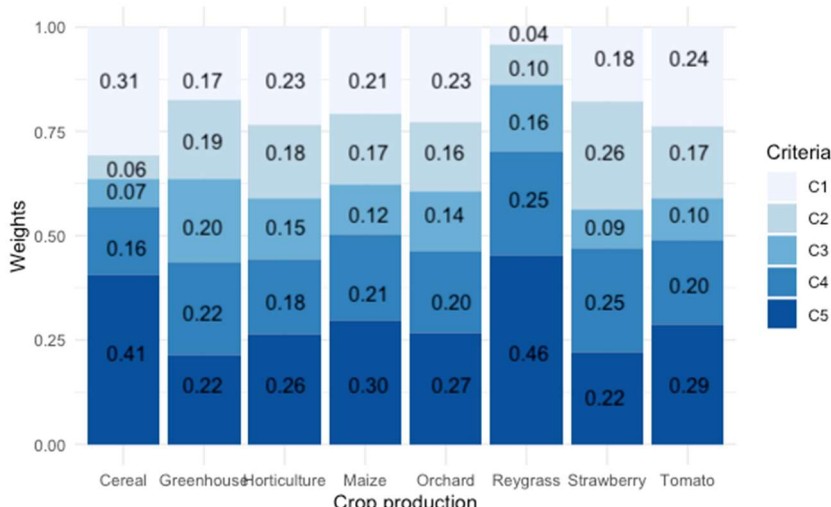

**Figure 4.** Weights of criteria grouped according to the crop production in Campania (IT). C1—Improve easy access to information; C2—Ensure coherent data and data reporting; C3—Improve delivery efficiency; C4—Improve public and private awareness; C5—Ensure economic sustainability.

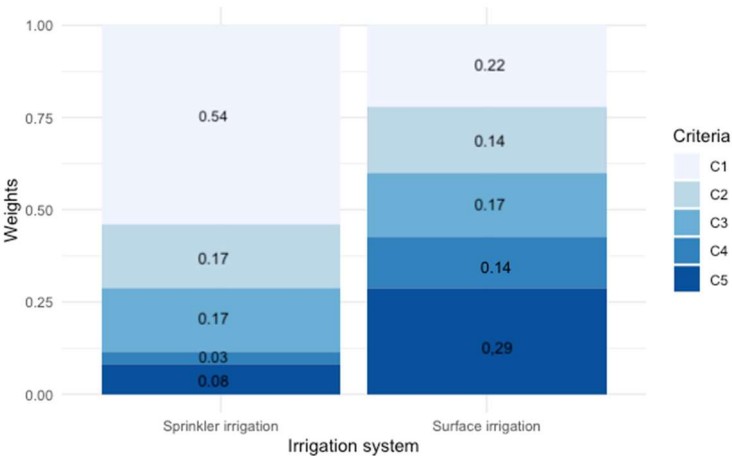

**Figure 5.** Weights of criteria grouped according to irrigation systems in Limburg (NL). C1—Improve easy access to information; C2—Ensure coherent data and data reporting; C3—Improve delivery efficiency; C4—Improve public and private awareness; C5—Ensure economic sustainability.

It appears evident that farmers who use this type of system are more interested in information concerning remote detection and weather forecasting. According to this profile of respondents, IASs will have to offer these services, which are of fundamental importance to the farmers.

Kujawsko-Pomorskie (PL): Results in Table 3 show that the stakeholders preferred C3—*Improve delivery efficiency*, which refers to the ability to ensure prompt and constant delivery of information to farmers. According to the results generated through the performed surveys, the farmers' expectations in the region are to obtain reliable information on the actual meteorological and soil moisture conditions. They also expect to know when, how much, and which crop should be irrigated. The above-mentioned need should be the main feature of the IAS. However, if we analyze the weights by grouping them according to the irrigation manager, it is evident that the most significant criterion for farmers who adopt drip and sprinkler irrigation is C5—*Assuring economic sustainability.* The long-term viability of drip irrigation also depends on its economic sustainability. Despite the potential increase

in water use efficiency and yield, the system also needs to generate higher income to be popular among farmers [4].

Andalucía (ES): The respondents from the Andalucia study site were all olive farmers. There were no farmers with diversification of production in the sample. The final ranking shows that, according to the stakeholders' judgments, C2—*Ensure coherent data and reporting are priorities*, was a criterion with a higher weight (Table 3).

## 5. Discussion

In the present study, there were a few limitations in the application of the AHP. It is worth noting that these priority rankings based on the collected data are, at times, quite flattened. As previously mentioned, one of the possible causes attributed to the homogeneity of the weights could be the inconsistencies in the matrices that express the farmers' judgments. The AHP has a means for measuring any inconsistencies by a formula called the consistency ratio. A ratio of 0 means a perfect consistency, while any ratio over 0.1 is considered inconsistent [48]. In the present work, only 26% of the subjects had a consistency ratio equal or lower than this limit. In this study, the inconsistency is mainly attributable to two aspects:

1. Method of structuring the model and criteria considered: Ideally, one would structure a complex decision through a hierarchy where factors at any level are comparable. If this condition does not occur during the criteria selection process, the possibility of generating inconsistencies among the elements of the pairwise comparison matrix (hinting at some randomness in the answers from the respondents) increases.
2. Method of administration of the questionnaire: It emerged that the mailed surveys made it difficult for respondents and researchers to interact. The letter was a necessary condition to explain the meaning of the pairwise comparison involved in the multi-criteria AHP analysis and to ensure that the respondents had full awareness and understanding of the criteria that they had to compare. It would have been appropriate to ask the interviewees to re-evaluate their judgments within the matrices, but this was not carried out because it would have been a difficult and time-consuming process.

However, it is evident from the results that there are further chances to improve the application of the AHP model for a better evaluation of the stakeholders' judgments. The literature offers an extremely broad overview of the advantages and disadvantages of the method on the consistency/inconsistency of the answers given by decision makers, and therefore, on the reliability of the model. Forman [49] introduced several comments related to the AHP. The most common reason for inconsistency is the lack of perfect knowledge. For this reason, it becomes essential to support the interviewee during the interview to clarify any doubts and reduce the possibility of error in their answers. In order to increase the use of IASs among farmers, it is necessary that these services acquire characteristics and performances that allow for "*Assuring economic sustainability*" (C5). If an investment is needed to improve irrigation management through the IASs, it will be justified by farmer users only if it is profitable. It would be interesting to extensively discuss the meaning of "*Assuring economic sustainability*" for a farmer, as it is a very complex condition that depends on many factors that are unpredictable, especially in the long term, because they could depend on future economic conditions. In a future project, the AHP could be a useful tool to deeply investigate these unpredictable factors in a qualitative and quantitative framework.

## 6. Conclusions

This paper addresses a very relevant issue framed in the process of the new CAP reform, which is the efficient and sustainable use of water, especially in the context of progressive scarcity of this resource in the Mediterranean area within and outside of the EU. This issue is a relevant commitment of the agricultural sector in the wider framework of the Agenda 2030 and the fight against the climate change, and there is a busy and rich research agenda ahead on this matter.

This work contributes to the general issue of water use in agriculture by developing a methodological approach based on the analytic hierarchy process to support the decision-making objective of "improving the use of irrigation advisory services". The results highlight that the most important criterion is C5—"Assuring economic sustainability", which means that the cost of IAS should be economically justified (i.e., economically affordable).

In order to enhance our comprehension of this topic, it is necessary to address the subsequent queries: What are the farmers' objective priorities (economic, non-economic, or both)? What are the tools employed by farmers to attain their aims, and what are their genuine objectives? Which factors exert an influence on them? The proposed reflections may be developed in future research activity, departing from the results of the present analysis. Finally, it is important to continue with this type of discussion to ensure that the decision-making process is able to contribute effectively to agricultural development in terms of sustainable irrigation management. Furthermore, using a database (the key information provided by the interviews) made it possible to aggregate the individual priorities in each study area and for each characteristic of the samples by relating the preferences expressed by the stakeholders.

Finally, the findings indicate that the criteria (C1, C2, C3, C4, C5) had varied forms of impact on the end users' judgments, and these attributes play a crucial role in shaping the strategies and scenarios for advancing the implementation of IASs.

**Author Contributions:** Conceptualization. I.I.M.D., F.A., A.D.M., T.D.G. and O.C.; methodology, I.I.M.D., F.A., A.D.M., T.D.G., O.C., Z.S., B.S. and D.V.; validation, F.A., O.C. and A.D.M.; investigation, F.A., T.D.G., O.C., A.D.M. and I.I.M.D.; data curation, I.I.M.D. and F.A.; writing—original draft preparation, I.I.M.D., F.A. and A.D.F.; writing—review and editing, A.D.F., A.M.; R.H. and I.I.M.D.; visualization, I.I.M.D.; supervision, F.A. and A.D.F.; project administration, F.A. and A.D.M.; funding acquisition, F.A. and A.D.M. All authors have read and agreed to the published version of the manuscript.

**Funding:** This research received WaterWorks2015 ERA-NET Cofunded Call (JPI Water) funding.

**Institutional Review Board Statement:** Not applicable.

**Data Availability Statement:** Data are available on request from the corresponding author.

**Acknowledgments:** This study was carried out as part of the OPERA project. The authors would like to thank the EU and The Ministry of Economic Affairs (The Netherlands), CDTI (Spain), MINECO (Spain), ANR (France), MIUR (Italy), NCBR (Poland) and WRC (South Africa) for funding, in the frame of the collaborative international consortium OPERA financed under the ERA-NET Cofund WaterWorks2015 Call. This ERA-NET is an integral part of the 2016 Joint Activities developed by the Water Challenges for a Changing World Joint Program Initiative (Water JPI).

**Conflicts of Interest:** The authors declare no conflict of interest. The partners of the OPERA project were informed and involved in the work, facilitating the access, and the use of data.

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
