# Peer review of "An Analysis of Preference Weights and Setting Priorities by Irrigation Advisory Services Users Based on the Analytic Hierarchy Process"

_agriculture, doi:10.3390/agriculture13081545_

Round 1

Reviewer 1 Report

When checked in Turnitin anti-plagiarism software, the manuscript has a degree of similarity of 38% (see the attached PDF file). There are phrases and paragraphs (for example subsections 2.1.1, 2.1.2, 2.1.3, 2.1.4 etc.) that appear to be from other bibliographic sources, and not quoted properly.

Source 1 from Turnitin (11%) not quoted properly: Francisco José Blanco-Velázquez & María Anaya-Romero- “Identifying sector needs to increase resource use efficiency“

http://opendata.waterjpi.eu/dataset/2a2a87e0-5c84-42cd-a9da-ecac0bbb9257/resource/09d7444c-c5e2-4473-835b-9c28f27d20d3/download/d1.1_report_stakeholder_opera.pdf

Source 2 from Turnitin (4%) with the same Title “Application of the Analytic Hierarchy Process in a Multi-Criteria Selection of Preferences Expressed by Stakeholders to Improve the Use of IAS”

I recommend that the authors re-phrase the phrases/ paragraphs that appear colored in the attached pdf file, or put them between quotation marks, with the bibliographic source immediately after them, to reduce the rather high percentage of similarity.

For figure 1, I recommend increasing the resolution as well as the fonts for better visualization by journal readers.

I recommend to the authors to enhance a little bit the Introduction section. Since there is no separate Literature Review section, and it is somehow condensed in the Introduction, I would suggest a more focused literature review that highlights the existing research on IASs and agricultural water management practices. Instead of mentioning various studies in a broad sense, I believe that perhaps it would be more helpful to summarize the key findings and limitations of previous research in this specific field.

I also suggest that the authors should highlight more on how the present study builds upon or contributes to the existing literature, and the novel insights (or methodologies) it offers.

I also suggest that that the Materials and Methods section could be developed some more, so to expand on the step-by-step explanation of how the pairwise comparison matrices were constructed, how the weights were calculated, and how the consistency ratio was determined.

I also recommend that the authors explain why they chose the AHP method for their experiment. Other methods (at least for their future studies) that could be considered, include: the Technique for Order of Preference by Similarity to Ideal Solution (TOPSIS) method for comparing alternatives based on their distance from the ideal solution, the Preference Ranking Organization Method for Enrichment Evaluations (PROMETHEE) method for ranking alternatives based on pairwise comparisons of criteria, and the Weighted Sum Model method for combining criteria in a multi-criteria decision-making problem by assigning weights to each criteria, Fuzzy logic that can be especially useful when dealing with uncertain or incomplete data, Neural networks that can be trained to recognize patterns and relationships within datasets etc.

In the Discussion section, I recommend that the authors discuss some more on the critical analysis of the results and compare with the existing literature. They could discuss the limitations of the AHP method, address the issue of inconsistencies in judgments, explore possible reasons for the variations in rankings etc. It would be interesting to read some more about how authors’ findings compare with other studies that have applied AHP in agriculture and water management.

I also suggest that the authors enhance the Conclusions, emphasizing some more the practical implications of the results and providing recommendations for policymakers and stakeholders (more that quoting other studies and publication). Additionally, I consider that the manuscript would benefit from clear suggestions for future (continuation) research.

Reviewer 2 Report

The author collected and studied data from four regions in the form of questionnaires. Through the analysis of the focus of chromatography analysis, it was found that due to the differences in crop planting and irrigation methods, farmers have different interests in IAS and hope for improvement and optimization. This provides reference and consideration points for the effective utilization of IAS. Overall, this study helps to some extent share and use IAS with farmers and application users, helping irrigation managers and users in various regions to improve yield and water resource utilization.However, some of the discussions, discussions, literature citations, and details in this study are problematic and urgently need to be addressed and corrected. The main comments are as follows:

1.Page 1, Line1. Title:

1) "the use of IAS" is not accurately described in the title; "use" is not enough to illustrate the point of this article.

2) It is not recommended to use a noun abbreviation in the title, such as "IAS".

2.Page 1, Line20. Abstract:

1) Research background is important for readers to understand this research.

2) It is recommended that the country chosen for the study be identified and linked to the actual case location.

3) Distribution and detailed distribution of questionnaires may be important. In addition, the crop type and irrigation method of farmers and users have a strong influence on the selection tendency.

3.Page 1, Line 37. Introduction:

1)Page 1, Line 38, as an introduction, should focus on describing the background, but in this study, naming the research content at the beginning is not a good way to write a paper.

2)Page 1, Line 38, The main research content of this paper may be different from that provided by the comprehensive method.

3) On Page 1, Line 38, the research language is suggested to be reviewed again, and some descriptions are not smooth and specific.

4) Line 87, part of what is described in this paragraph is repeated. In addition, the information provided in Table 1 seems to have no impact on the development of this study.

5) Line 98, readers prefer to see the content of the research and the extension and contribution to the field of research in the concluding paragraph, rather than the brief title of each section.

4.Page 3, Line 105. Materials and Methods:

1)Page 3, Line 107, where there is ambiguity, is it "model" or "method"? The "Analytic hierarchy process model used" should be followed by the type of data that can be analyzed.

2)Page 3, Line 111. The description of the region in this part is over-discussed, and some information is missing. For example, the description of olives uses too many words; The specific assignment for the questionnaire is not appropriate to list here. In addition, the case selection seems to have different priorities, but it is not clearly stated.

3)Page 6, Line 175, if the first sentence is superfluous, perhaps it should be put in the previous paragraph. In addition, the research method and the questionnaire content can be further separated, so that it will look more smooth.

4)Page 6, Line 191, the background and research introduction of analytic hierarchy process are suggested in the introduction.

5)Page 7, Line 210, does not explain the meaning of the letters in the formula.

6)Page 7, Line 229, the content of standardization and consistency tests intersects and is difficult to read fluently, and the methods should be interpreted sequentially.

7)Page 8, Line 243, if the calculation flow is written, there is no need to describe it again.

5.Page 8, Line 247. The Results:

1)Page 8, Line 248, the author should re-examine the language description of this part, which is difficult to understand. In addition, important conclusions are suggested after the analysis, there is a process of introduction.

2)Page 10, Line 291, the content of this paragraph is obtained from the analysis of Figure 3, and should be presented in the first sentence. In addition, the results of Table 5 are not described there.

3)Page 11, Line 312, In the results, there is a difference between drip irrigation and spray irrigation in Poland, but the relevant content is not found in the chart.

4)Page 11, Line 324. The research area in this paper is defined as part of the country, but the name of the country is used in the results, whether there is an expanded scope.

6.Page 11, Line 328. Discussion:

1)Page 11, Line 329, there is ambiguity in the first sentence, the author is advised to thoroughly check the language of this article.

2)Page 11, Line 351, the author does not recommend using an absolutist sense when describing analytic hierarchy Process. In addition, the author should try to explain the advantages and functions of analytic hierarchy process, rather than explain the application of analytic hierarchy process.

3)Page 11, Line 361, although the author introduces the discussion of the wrong judgment of the analytic hierarchy process, the seeming consistency of the content is too much, so that it loses its focus.

7.Page 12, Line 389. Conclusions:

1)Page 12, Line 390, it is not recommended to cite literature in the conclusion part.

2)Page 12, Line 405, content cites too much and lacks necessary key conclusions.

8.Page 13, Line 423. References:

1) Incorrect literature citations. For example, [55], [4], etc.

2) There are a lot of literatures in the analytic hierarchy process, but it does not highlight the key points in the paper. It is suggested that the author summarize the literatures and decide whether to cite them or not.

3) There are repeated references. Examples include [27] and [52].

4) Some documents are not in standard format.

5) It is recommended that authors proofread literature carefully to reduce unnecessary references.

Moderate editing of English language is required

Reviewer 3 Report

Introduction: The introduction lacks clarity on the purpose of improving the use of Irrigation Advisory Services (IASs). The context also fails to adequately explain the significance of this improvement and why it is essential to utilize the Analytic Hierarchy Process (AHP) to fully understand it.

Check space issue in the sentence, for example page 2, Line 78.

Methodology: How the AHP criteria developed?

In general, I find the utilization of IASs in various contexts to be quite perplexing. It necessitates a clear explanation and understanding of how to incorporate it into your research. Can you provide more information on this topic?

Page 6 Line 187, Transdisciplinary framework, what is this?

Page 11, Line 32-342: the whole paragraph could be part of the introduction or somewhere else where author describes about AHP.

Discussion: The work requires significant improvements, particularly in the authors' interpretation of the findings, research problem, and existing literature. I am interested in understanding how the findings contribute to the relevant knowledge base and their practical and policy implications.

Please check for formating and sentence structure.

Round 2

Reviewer 1 Report

Still a similarity percentage of 29% in Turnitin, with again entire paragraphs copied from other sources (see the pdf file), especially sections 3.1.1-3.1.4.

( I do not understand why the authors keep insisting on those subsections/ paragraphs, still using them, without rephrasing or putting into quotes, and not quoting them correctly, if the paragraphs/sections are not their own work...)

11% similarity  from "Identifying sector needs to increase resource use efficiency",
 Authors Francisco José Blanco-Velázquez & María Anaya-Romero

http://opendata.waterjpi.eu/dataset/2a2a87e0-5c84-42cd-a9da-ecac0bbb9257/resource/09d7444c-c5e2-4473-835b-9c28f27d20d3/download/d1.1_report_stakeholder_opera.pdf?cv=1

Reviewer 2 Report

The authors have addressed all my concerns in the modified manuscript. I think it could be accepted by the journal. 

Author Response

Reviewer's comment has been received and we are very thankful for the constructive comments and feedback .